# Two-Lane DNN Equalizer Using Balanced Random-Oversampling for W-Band PS-16QAM RoF Delivery over 4.6 km

**DOI:** 10.3390/s23104618

**Published:** 2023-05-10

**Authors:** Sicong Xu, Bohan Sang, Lingchuan Zeng, Li Zhao

**Affiliations:** 1Shanghai Institute for Advanced Communication and Data Science, Key Laboratory for Information Science of Electromagnetic Waves (MoE), Fudan University, Shanghai 200433, China; 2Aerospace Information Research Institute, Chinese Academy of Sciences, Beijing 100094, China

**Keywords:** deep neural network, photonic-aided mm-wave system, coherent detection DSP

## Abstract

For W-band long-range mm-wave wireless transmission systems, nonlinearity issues resulting from photoelectric devices, optical fibers, and wireless power amplifiers can be handled by deep learning equalization algorithms. In addition, the PS technique is considered an effective measure to further increase the capacity of the modulation-constraint channel. However, since the probabilistic distribution of m-QAM varies with the amplitude, there have been difficulties in learning valuable information from the minority class. This limits the benefit of nonlinear equalization. To overcome the imbalanced machine learning problem, we propose a novel two-lane DNN (TLD) equalizer using the random oversampling (ROS) technique in this paper. The combination of PS at the transmitter and ROS at the receiver improved the overall performance of the W-band wireless transmission system, and our 4.6-km ROF delivery experiment verified its effectiveness for the W-band mm-wave PS-16QAM system. Based on our proposed equalization scheme, we achieved single-channel 10-Gbaud W-band PS-16QAM wireless transmission over a 100 m optical fiber link and a 4.6 km wireless air-free distance. The results show that compared with the typical TLD without ROS, the TLD-ROS can improve the receiver‘s sensitivity by 1 dB. Furthermore, a reduction of 45.6% in complexity was achieved, and we were able to reduce training samples by 15.5%. Considering the actual wireless physical layer and its requirements, there is much to be gained from the joint use of deep learning and balanced data pre-processing techniques.

## 1. Introduction

With the advancement of 5G mobile communication, millimeter band systems can provide higher data rates and less wireless interference, even in high-mobility environments [1,2,3,4,5,6,7,8,9,10,11]. Moreover, as wireless communications develop, the wider bandwidth expansion is driven by the growing demand for capacity in next-generation 6G mobile networks, which is expected to lead to ultra-high data rates [12,13]. Recently, W-band (75–110 GHz) has received more and more attention, with its wide bandwidth, relatively low atmospheric loss, and good directionality. It has been considered as a potential candidate for long-distance mobile communication [14,15,16,17]. Photonics-aided W-band signal generation is a critical form of technology in long-distance radio-over-fiber (ROF) transmission systems. It breaks the bottleneck of the bandwidth limitation of traditional electrical equipment and can achieve cost-effective and stable high-frequency millimeter-wave signals [18,19,20,21].

To further enlarge the capacity of the modulation-constraint channel, the probabilistic shaping (PS) technique is introduced into optical fiber communication systems [22]. Moreover, with the development of the PS technique, significant breakthroughs can be achieved in expanding transmission distance and improving spectral efficiency [23]. However, simultaneously, photoelectric and electronic devices usually introduce nonlinear impairment. This impairment is present not only in the optical fiber system, but also in the photonics-aided mm-wave ROF system [24], which is a significant constraint factor of W-band ROF transmission systems. In most of the recent research, neural networks (NN) and deep learning [25,26,27,28,29,30,31,32,33,34,35,36,37] have been proven to be one of the most effective equalizers for RoF delivery. Recently, we [38] designed a dual gated recurrent unit neural network-based nonlinear equalizer (dual-GRU NLE) in the W-band mm-wave transmission over a 10 km SMF (single mode fiber) and a 1.2 m wireless link. It has been proven that two-lane DNN equalizers made up of two-lane independent real-valued DNNs dealing with in-phase and quadrature component training data have superiority in nonlinear compensation, and the training of paradigms in multi-classification usually employs a real-valued (RV) input, an RV active function, and RV weights by discarding the inherent phase content [39,40]. However, overly complex models, including too many parameters, can lead to over-fitting suppression. Furthermore, introduction of the PS algorithm can cause a severe skew in the generalization of the PS-16QAM data. Such a bias in the training dataset may result in a machine learning imbalance, leading to inadequate training of the minority class. Several studies have shown that this problem can be overcome by randomly resampling the training dataset, which can be achieved by undersampling or oversampling [41]. Random undersampling deletes the length of the training dataset from the majority class, and, thus, may lead to the loss of useful information, which further results in capacity reduction. In contrast, random oversampling (ROS) is prone to increasing the training dataset of the minority class, which has been demonstrated to be an effective way to overcome machine learning imbalance.

In this paper, we demonstrate a two-lane DNN (TLD) equalizer combined with ROS (TLD-ROS) for the first time. It is adaptive to the PS technique, which is especially employed in long-distance ROF transmission systems. We experimentally realized a single-channel 10-Gbaud W-band PS-16QAM transmission over a 100 m optical link and a 4.6 km wireless link. In terms of the receiver sensitivity, the PS-16QAM signal using TLD-ROSQAM equalizers showed a 1 dB gain over typical NN equalizers without balancing ROS when the BER reached the HD-FEC (hard decision-forward error correction) threshold of 3.8 × 10^−3^. Compared with the traditional nonlinear DNN equalizer and Volterra-series, it showed a better performance in terms of good training accuracy, less training size demand, and computation complexity.

## 2. Principle

### 2.1. Random Oversampling (ROS)

For the NN classification algorithm, a class imbalance occurs when the number of samples in some classes is greater than that of instances in other classes. Here, we define the larger data set as the majority class and the smaller one as the minority class. For long-range high-speed wireless channels influenced by nonlinearity, phase noises, and high loss, it is difficult to learn from the samples of the minority class in the presence of serious class overlapping. The random oversampling (ROS) algorithm’s main function is to overcome the imbalance problem by redistributing the training dataset. We employed the ROS technique on the Rx side of the W-band PS-16QAM RoF delivery in our work. The basic principle of ROS is shown in Figure 1. As we know, N-order modulation signal equalization is regarded as N-classification. The PS technique introduces a severe skew in the class distribution of the baseband PS-16QAM data, causing the generation of the majority class in the inner rings and a few minority classes in the outer rings, which can result in the original imbalanced dataset, as shown in Figure 1a. The core idea of the ROS algorithm is to randomly extract samples from the minority classes and make multiple replications in order to balance the class distribution of the training set [42,43]. The procedure using the classic ROS algorithm to generate a training set $\left\{S^{\prime }\right\}$ is described as follows:

Step 1: For an original training dataset $\left\{S\right\}$, the data size in the smallest minority class is counted as $N^{-}$. Meanwhile, the data size in the largest majority class is written as $N^{+}$, and the sample rate is *SR*. Here, $N^{+}+N^{-}=N,$ and $N+N^{-}\times SR$ is the length of the final training dataset for every classification. The imbalance ratio is $\lambda =\frac{N^{+}}{N^{-}}$. 

Step 2: For one specific minority classification and the largest majority classification, all of their original samples constructed the original minority class training set $\left\{S^{-}\right\}$ and majority class training set $\left\{S^{+}\right\}$, respectively.

Step 3: For $i=1:N^{-}\times SR$, we randomly chose numbers from $\left[1,N^{-}\right]$, and find the corresponding sample set $\left\{x^{\prime }\right\}$ from $\left\{S^{-}\right\}$.

Step 4: The selected samples were added to the minority class set $\left\{S^{-}\right\}=\left\{S^{-}\right\}+\left\{x^{\prime }\right\}$.

Step 5: The generated training set was obtained $\left\{S^{\prime }\right\}=\left\{S^{+}\right\}\cup \left\{S^{-}\right\}$.

Step 6: For all minority classifications, we repeated the operations above and obtained a large number of samples. In addition, for majority classifications, we extracted only a small subset of samples from the imbalanced dataset. Finally, we obtained balanced dataset generation.

### 2.2. Two-Lane DNN (TLD) Equalizer

A two-lane DNN (TLD) equalizer was employed in our experiment to mitigate the nonlinear impairment. The so-called “two-lane DNN” represents that such an equalizer is composed of two DNNs, one designed to process real signal sequences and the other applied for imaginary signal sequences, since our modulated signal is a complex 16QAM signal. Figure 2 shows the schematic diagram of the TLD equalizer combined with ROS. 

In the training process, our proposed TLD equalizer was trained via two steps. In the first step, the training dataset was randomly oversampled to generate balanced 16QAM signals. In the second step, the equalizer was trained with the help of the training-balanced 16QAM sequence, and the weight value of the TLD equalizer was further optimized until the target error value was achieved. In the testing process, in contrast, 30% of the originally received PS-16QAM signal used as the testing dataset was directly inputted into the well-trained TLD neural network. Finally, the BER decision could be implemented according to the equalized testing signals. Therefore, it is worth noting that ROS was only implemented into the training data, and the TLD network was trained well with the help of balanced training data. 

In general, a NN is made up of an input layer, several hidden layers, and an output layer. Our proposed TLD equalizer has one input layer, *L* (set as two in our experiment) hidden layers, and one output layer. Considering the complexity of a typical M-layer NN described as $C_{MLP}=n_{0}n_{1}+n_{1}n_{2}+n_{2}n_{M}$ [44], the neural network we used was a simple and low-layer one, with a structure of [180-300-300-1]. There were 180 units in the input layer, 300 cells for both hidden layers, and one unit for output. Such parameters are proven to be capable of satisfying the need for nonlinear equalization without overly increasing the complexity of the neural network in the following sections. $w_{I,mj}{}^{k}$ and $w_{Q,mj}{}^{k}$ are the weight values of two lanes, respectively, which link the hidden layers and the output layer, where *I* or *Q* denotes the *I* or *Q* lane of signals it deals with, respectively. *k* represents the current *k-th* layer, (*0-th* for the input layer, and *3-th* for the output layer), and *m* and *j* represent the *m-th* node in the former hidden layer and the *j-th* node in the current layer, respectively. Firstly, the weight value randomization ($w_{I,mj}{}^{k}$ and $w_{Q,mj}{}^{k}\in \left[-1,1\right]$) initialized their TLD equalizers, learning rates, and iterative learning epoch settings. Secondly, the oversampled PS-16QAM dataset was separated into *I* and *Q* two-lane vectors, respectively, and then sent to the input layer with a length of $n_{0}$. As shown in Figure 2, the input vectors can be written as $X_{I}\left(n\right)=\left[x_{I}\left(n\right),x_{I}\left(n-1\right)\ldots ,x_{I}\left(n-n_{0}+1\right)\right]$ and $X_{Q}\left(n\right)=\left[x_{Q}\left(n\right),x_{Q}\left(n-1\right)\ldots ,x_{Q}\left(n-n_{0}+1\right)\right]$, and then multiplied by weight values $w_{I,mj}{}^{1}$ and $w_{Q,mj}{}^{1}$ in the first hidden layer, respectively. Because the neurons between the layers are fully connected, the output of the *j-th* neurons in the first hidden layer can be described as follows: (1)$$h^{1}{}_{I,j}=f(\overset{n_{0}}{\underset{m=1}{\sum }}w^{1}{}_{I,mj}\;x_{I,m)}$$
(2)$$h^{1}{}_{Q,j}=f(\overset{n_{0}}{\underset{m=1}{\sum }}w^{1}{}_{Q,mj}\;x_{Q,m)}$$
where $n_{i}$ denotes the number of nodes in the *i-th* layer, with $n_{0}$ and $n_{M}$ defining the numbers of nodes in the input layer and output layer, respectively. Based on the feed-forward training process, the output of the *l-th* neuron in the 2-*th* hidden layer can be calculated as follows:(3)$$h^{L}{}_{I,l}=f(\overset{n_{1}}{\underset{m=1}{\sum }}w^{L}{}_{I,ml}\cdot h^{L-1}{}_{I,m})$$
(4)$$h^{L}{}_{Q,l}=f(\overset{n_{1}}{\underset{m=1}{\sum }}w^{L}{}_{Q,ml}\cdot h^{L-1}{}_{Q,m})$$

Here, as mentioned above, *L* is set to 2, and $n_{1}\;$ is the number of nodes in the *1-th* hidden layer. Take $w^{L}{}_{I,ml}$ in the DNN dealing with *I*-lane signals as an example: it represents the weight value in the *2-th* hidden layer of the *m-th* node in the *L-1-th* hidden layer and the *l-th* node in the current layer. $h^{L}{}_{I,m}$ and $h^{L}{}_{Q,m}$ denote the output of the *m-th* neuron in the *L-th* hidden layer for the *I*-lane DNN and *Q*-lane DNN, respectively. It is worth noting that *f *(*.*)** denotes the nonlinear active function between hidden layers. In DNN, there are several common active functions, such as “sigmoid” and “tanh” functions. Taking gradient explosion and gradient vanishing into consideration, we chose the “ReLu” function, which can be described as: (5)$$f\left(x\right)=relu\left(x\right)=\max\left(0,x\right)$$

Meanwhile, the transformation from the hidden layer to the output layer is linear, contrary to that from the input layer to the hidden layer, which is nonlinear. Considering this, we thus chose the “purelin” function as the active function *g *(*.*)** in the output layer. The final equalized output result is given as:(6)$$O_{n}=g(\overset{n_{2}}{\underset{m=1}{\sum }}w^{L+1}{}_{I,mj}h^{L}{}_{I,m})+i\cdot g(\overset{n_{2}}{\underset{m=1}{\sum }}w^{L+1}{}_{Q,mj}h^{L}{}_{Q,m})$$

We noted that the weight values adaptively updated based on the least mean squares (LMS) error function, which can be given as:(7)$$J_{n}=\frac{1}{2}\overset{T}{\underset{n=1}{\sum }}{\left(T_{n}-O_{n}\right)}^{2}$$
where $J_{n}$ is the cost function and *T* refers to the length of the training dataset $X\left(n\right)$. Then, we subtracted the predetermined expected output value $T_{N}$ from the obtained output result $O_{n}$ to obtain the error value $e_{n},$ which was fed back to TLD equalizers to participate in the calculation. With the aid of BP algorithms, the weight values $w_{I,mj}{}^{k}$ and $w_{Q,mj}{}^{k}$ were updated constantly until reaching the preset epoch or error value. The process of iterations can be given as:(8)$$w^{k+1}{}_{I,mj}=w^{k}{}_{I,mj}-\Delta w^{k}{}_{I,mj}=w^{k}{}_{I,mj}-\eta \frac{\partial real\left(Jn\right)}{w^{k}{}_{I,mj}}$$
(9)$$\frac{\partial real\left(J_{n}\right)}{\partial w_{I,mj}{}^{k}}=real\left(e_{n}\right)\cdot \frac{\partial real\left(O_{n}\right)}{\partial w_{I,mj}{}^{k}}\cdot \frac{\partial h_{I,j}{}^{L}}{\partial h_{I,j}{}^{L-1}}\ldots \frac{\partial h_{I,j}{}^{k}}{\partial w_{I,mj}{}^{k}}$$
(10)$$w^{k+1}{}_{Q,mj}=w^{k}{}_{Q,mj}-\Delta w^{k}{}_{Q,mj}=w^{k}{}_{Q,mj}-\eta \frac{\partial imag\left(Jn\right)}{w^{k}{}_{Q,mj}}$$
(11)$$\frac{\partial imag\left(J_{n}\right)}{\partial w_{Q,mj}{}^{k}}=imag\left(e_{n}\right)\cdot \frac{\partial imag\left(O_{n}\right)}{\partial w_{Q,mj}{}^{k}}\cdot \frac{\partial h_{Q,j}{}^{L}}{\partial h_{Q,j}{}^{L-1}}\ldots \frac{\partial h_{Q,j}{}^{k}}{\partial w_{Q,mj}{}^{k}}$$

Here, η denotes the learning rate and the Δ  symbol refers to calculations of the gradients. In our proposed TLD-ROS equalization scheme, the original received PS-16QAM signal was firstly processed by deployed ROS, and then the target output was corresponding modified. Finally, the average distributed 16QAM was placed into the TLD classifier to obtain the optimum network parameters.

## 3. Experimental Setup

Figure 3 shows the experimental setup of our presented W-band PS-16QAM delivery over a 4.6 km free space wireless transmission system. As the transmitter side shown in Figure 3a, an 88.5 GHz frequency space was generated by two individual external cavity lasers (ECLs) based on optical heterodyne beating. The baseband PS-16QAM signal was firstly generated via offline Matlab-2020b software, and then digital-to-analog converted (DAC) by the arbitrary wave generator (AWG) with a sampling rate of 100 GSa/s. Boosted by two parallel electrical amplifiers (EAs), the baseband *I* and *Q* component of the PS-16QAM signal created by the Pseudo-Random Binary Sequence (PRBS), with a length of $2^{20}$−1, was used to drive the *I/Q* modulator. In order to avoid overfitting in the equalization process, we used the PRBS of order 20 to generate the transmitted sequence. The periodic length of a PRBS-20 is 1048575 (2^20^ −1 = 1,048,575 bits). The total transmitted *I/Q* length extracted from the generated PRBS-20 sequence was 163,84 (16,384 symbols = 65,536 bits), much less than 1,048,575, and effectively avoided the repetition of pseudo-random numbers. The external cavity laser (ECL1) at 1550 nm, with a line width of 100 kHz and an average power of 16 dBm, was modulated as a signal light source to carry transmitted PS-16QAM data by I/Q modulator with a 3 dB optical bandwidth of 30 GHz and a half-wave voltage of 2.7 V at 1 GHz. Then, the optical beam was amplified by a polarization-maintaining erbium-doped fiber amplifier (PM-EDFA). Then, the ECL2 at the center wavelength of 1549.3 nm, operating as a local oscillator (LO) with a frequency space of 88.5 GHz with the ECL1 light, was used to generate W-band mm-wave signals. After being delivered over a 100 m standard single mode fiber (SSMF), the optical power of the combined beam was adjusted by an attenuator (ATT) to obtain the optimum input power into the photodiode (PD), and PD was implemented within the frequency range of 10∼170 GHz at −2 V DC bias and with an output power of −7 dBm.

As depicted in Figure 3a, at the transmitter, the output 88.5 GHz PS-16QAM signal was amplified by a low-noise amplifier (LNA) with a 30 dB gain and a cascaded power amplifier (PA, saturated output 18 dBm), and then the boosted W-band signal was fed into the W-band horn antenna (HA) with a 30 dBi gain. A pair of identical lenses (Lens 1 and Lens 2), the diameter and focal length if which were 10 cm and 60 cm, respectively, was designed to amplify mm-wave signals, thereby improving SNR. The pair of lenses was placed just between the transmitted HAs to focus the wireless mm-wave signal at the position of the received W-band HA. 

Figure 4 shows the photos of the long-distance radio-over-fiber delivery of the PS-16QAM signal over 4.6 km on Fudan campus. The Tx-side of the W-band 4.6 km wireless transmission system was situated at the Guanghua Building on the Handan Campus of Fudan University, and the Rx-side was located at the WuLi Building on the Jiangwan Campus of Fudan University. The distance between them was 4.6km on the satellite map. At the receiver, the received signal by HA was firstly boosted by a low-noise amplifier (LNA) with a 30 dB gain, and then down-converted into a 13.5 GHz intermediate frequency (IF) signal via a commercial mixer driven by a radio frequency (RF) signal with a frequency of 75 GHz. Additionally, the amplified IF signal after an EA with a gain of 26 dB was finally captured by a digital storage oscilloscope with a sampling rate of 100 GSa/s and an electrical bandwidth of 45 GHz. As shown in Figure 4, the offline DSP at the Rx-side is followed by down-conversion, resampling, the Gram–Schmidt orthogonalization procedure (GSOP), constant modulus algorithm (CMA), frequency offset estimation (FOE), carrier phase estimation (CPE), decision-directed–least mean squares (DD-LMS), and nonlinear equalization. Here, the nonlinear equalization methods which we compared include typical Volterra series, TLD, and TLD equalizers combined with ROS (TLD-ROS).

## 4. Experimental Results and Discussions

### 4.1. Traditional DSP Algorithm

Figure 5(a1–a5) provide the density distribution of PS-16QAM constellation diagrams after successive DSP algorithms, such as GSOP, CMA, FOE, CPR, and DD-LMS. As seen in Figure 5(a2), the constellation diagrams were circles after the CMA algorithm, but after the FOE algorithms, the frequency offset noise could be mitigated. In addition, by employing carrier phase recovery (CPR), the problem of phase offset was solved. After the DD-LMS algorithm, four points in the inner circle could be identified from complex-valued fuzzy signals. However, the remaining constellation points were widely scattered, as shown in Figure 5(a5). This implies that noises with a higher SNR requirement easily affect the signals in the outer circles. Meanwhile, in Figure 5(b1–b5), the complex-value 16QAM signals are successfully separated into real and imaginary parts after the aforementioned DSP algorithms. However, they are still fuzzy, since these steps are linear algorithms and are useless for nonlinear noises. In the 16QAM constellation diagram in Figure 5(b5), the signals recovered after DD-LMS are evenly distributed and relatively concentrated around the 16 points. This proves that PS-16QAM does not have the same density distribution as 16QAM, as PS-16QAM has a severe skew in the class distribution of constellation diagrams due to the PS algorithm. In order to conquer the uneven distribution problem, a pre-processing data method of ROS is considered a simple and valid approach to achieving balanced learning.

### 4.2. Results of ROS Processing

As seen in Figure 6a, the red-brown columns mark the amount distribution of constellation points in the inner ring of PS-16QAM, the blue ones in the middle ring, and the purple ones in the outer ring. The distribution of the transmitted PS-16QAM showed an uneven change with the information entropy. Such an imbalance becomes severe when the information entropy grows smaller. The process of the ROS method is presented here. The information entropy in our experiment was set as 5.6 bit/symbol. First, we oversampled the original unbalanced PS-16QAM training dataset. Thus, the corresponding target output has an equal distribution in Figure 6b. The constellation diagrams of PS-16QAM before ROS could not be fully separated into 16 points in Figure 6c, while in Figure 6d, it is shown that they could be divided into 16 equal parts after ROS.

There are two methods of ROS. In the first approach, called “ROS_I/Q_” in Figure 7a, the complex QAM sequence is firstly divided into the real-valued (RV) *I* and *Q* parts and then randomly oversampled separately. The *I* and *Q* components, after ROS_I/Q_, are finally trained through the TLD equalizer, respectively. In the other approach, denoted as “ROS_QAM_” in Figure 7b, the density of the complex QAM signal is directly flattened via the complex-valued ROS and then split into real and imaginary parts as the input dataset is sent into the two independent DNN equalizers, respectively. In our experiment, we assumed that 70% of the received symbols after DD-LMS were constructed as training samples. Therefore, when transmitting 16,384 symbols, the length of the training dataset was established to be around 11,000 (16,384×70%≈11,000), and the length of the testing set was around 5000 (16,384×30%≈5000). For the ROS_I/Q_ scheme, the sizes of the generated balanced training datasets for the I-branch and Q-branch were 17,208 and 17,108, respectively, caused by the difference between the *I*- and *Q*-distribution. For the ROS_QAM_ scheme, assuming that the lengths of the original and repeated samples were 11,000 and 15,864 symbols, respectively, the same target output number for the *I*-branch and *Q*-branch was 26,864 symbols (11,000 + 15,864 = 26,864).

Then, the category values Y (∈ [1,16]) were converted to standard complex 16QAM signals with the *I* data (∈ [−3,−1,1,3]) and *Q* data (∈ [−3,−1,1,3]). Furthermore, the generated ROS samples were fed into TLD with MSE loss, where the weight values on the *I-* and *Q*-paths, respectively, were updated until the threshold error value or iteration number was reached. 

### 4.3. BER Performances Using TLD Equalizer with or without ROS Preprocessing

Moreover, we compared the BER performance versus the intermediate frequency (IF) after down-conversion for 10-Gbaud PS-16QAM signals when the optical power into the uni-traveling-carrier photodiode (UTC-PD) was fixed as −1 dBm in Figure 8. It can be found that the BER increased with increasing IF, and when the IF was 21 GHz, there was steep fading in the high-frequency part of the electrical spectrum for the PS-16QAM signal, which severely affected the BER performance. In our experimental setup, the IF was optimized as 14 GHz. Meanwhile, we compared the BER performances of PS-16QAM signals, employing only TLD equalizers and the two different TLD-ROS equalizer schemes. It is obvious that the TLD-ROS equalizer schemes offer great help to the drop in BER compared with the TLD equalizer. Moreover, it is worth noting that in this case, the PS-16QAM signal was firstly shaped into a “flat” 16-QAM constellation and then trained by the neural network model, where a “perfect” ROS was carried out, namely, equiprobable symbols without repetition.

Figure 9a illustrates the BER performance versus the input optical power for the PS-16QAM signal and the 16QAM signal, respectively. All nonlinear NN equalizers have the same neural structure [180-300-300-1], where 180 defines the memory length of the input layer. It is evident that when employing traditional DSP algorithms, PS-16QAM has a better BER performance than 16QAM, since it has a lower demand for SNR. However, linear CMMA and DD-LMS DSP algorithms are ineffective against the nonlinear problem. On the contrary, with the aid of TLD equalizers, the BER performances improve significantly. There is a difference in BER between 16QAM signals and PS-16QAM signals employing TLD equalizers. When the optical input power is less than −1 dBm, the BER performance of PS-16QAM is better than that of 16QAM due to the use of TLD equalizers, because PS-16QAM is better adapted to channel transmission when the SNR is low. Contrarily, when the optical input power is larger than −1 dBm, 16QAM outperforms PS-16QAM, because the greater input power meets the higher SNR requirement of 16QAM. Moreover, the outer ring constellation points of PS-16QAM signals are minority classes, which suffers inadequate training during the machine learning phase. Therefore, PS-16QAM may not perform as well as 16QAM when the optical input power is greater than −1 dBm. It is essential to deploy the ROS technique to overcome the imbalance learning issue of PS-16QAM. With the help of ROS, the receiver sensitivity of PS-16QAM signals equalized by TLD-ROS was improved, as 1 dB at a BER of the FEC-HD threshold of 3.8 × 10^−3^ was compared with 16QAM signals only employing the TLD equalizers.

### 4.4. BER Performance Comparison Using Different Equalizer Schemes

#### 4.4.1. Training Accuracy

For the implementation of DSP, followed by carrier phase recovery, we deployed different schemes of various linear and nonlinear equalizers, including a linear 21-tap CMMA equalizer combined with 223-tap DD-LMS; a 321-2nd tap Volterra equalizer; a TLD equalizer with the same structure as [180-300-300-1], with 180 units in the input layer, 300 cells for both hidden layers, and one neuron in the output layer; and a TLD equalizer combined with an ROS balancing method when the transmission speed of the PS-16QAM signal was 10-Gbaud. As shown in Figure 10a, it can be found that the BER performance improved with greater optical power. We can see clearly from Figure 10b–d that with the increase in the optical power of UTC-PD, the recovered constellation diagrams after DD-LMS became more and more clear, although they still could not be separated into 16 parts. When fixing the input optical power at 0 dBm, we found that the recovered constellation diagrams after the Volterra equalizer were still fuzzy and focused on four areas. TLD equalizers performed better than the Volterra series in terms of learning accuracy because they helped constellation points to separate into 16 parts and demonstrated better BER performance. The BER further dropped when we employed our proposed TLD-ROS, and the constellation diagram became clearer. Moreover, it is worth noting that in this case, the PS-16QAM signal was firstly shaped into a “flat” 16-QAM constellation and then trained by the neural network model, where a “perfect” ROS was carried out, i.e., equiprobable symbols without repetition. Moreover, we compared two TLD-ROS schemes. As can be seen from the BER result in Figure 9a, we came to the conclusion that the TLD-ROSQAM scheme performs better than the former one, with a relatively lower BER. It was also is verified that ROS is an effective algorithm for handling the imbalanced learning problem. In particular, ROSQAM is more suitable for handling imbalances caused by PAS. In terms of the receiver sensitivity, the PS-16QAM signal using TLD-ROS_QAM_ equalizers showed a 1dB gain over typical TLD equalizers without balancing ROS, at a BER of 3.8 × 10^−3^.

#### 4.4.2. Training Dataset Size

Next, we compared the BER curves of PS-16QAM versus different training sizes of TLD schemes in Figure 11 when the optical input power was fixed at −1 dBm. It can be concluded that a larger training dataset is beneficial to BER reduction, although at the expense of efficient system capacity. Moreover, it was obvious that the BER of PS-16QAM employing the TLD-ROSQAM equalizer performed the best, and it was reduced to below 3.8 × 10^−3^ when the length of the training set exceeded 11,000. In contrast, the TLD-ROS_I/Q_ equalizer required 1000 extra training samples, and 2000 extra training samples for the TLD equalizer. This means the ROS method is not only helpful for the adequate training of minority categories, but is also superior in training sample reduction. Although the training size was increased based on ROS, the demand for training capacity was decreased. In the following discussion, the training dataset size was set as 11,000 samples.

#### 4.4.3. Memory Size in Input Layer

Furthermore, we compared the BER performance of PS-16QAM signals with different optical input powers versus memory sizes of the NN model. The BER performance significantly improved with the increase in the system’s SNR. As seen in Figure 12, it is evident that the HD-FEC threshold of 3.8 × 10^−3^ could not be met for PS-16QAM signals when the system’s SNR was low, with −3 dBm optical input power, until the memory size of the NN model exceeded 220. The HD-FEC threshold was reached with a memory size of only 180 when the optical input power was −1 dBm. However, BER performances improved with the increase in memory size despite the expense of computational complexity. Additionally, when the memory size was more than 180, the BER curve decreased slowly, since the overfitting effect occurred when the parameters of NN were too complicated. In light of the results mentioned above, 180 is the experiment’s preferred memory size for the NN model.

#### 4.4.4. Neuron Number in Hidden Layers

Moreover, we further compare the BER performance with an input optical power of −1 dBm versus the quantities of neural cells *n*_1_ and *n*_2_ in hidden layer 1 and hidden layer 2, respectively. The quantities of neurons *n*_0_ and $n_{M}$ in the input layer and the output layer were 180 and 1, respectively, for all TLD schemes. Here, the green dots mark the points at which the BER is 3.8 × 10^−3^. As shown in Figure 13, for TLD equalizers, the minimum data pair is 410 and 280. For TLD-ROS_I/Q_ equalizers, the minimum data pairs are 280–240 or 320–200, and 270–200 for TLD-ROS_QAM_ equalizers. Horizontally viewed from the axis in Figure 13, we can draw from the BER curved surfaces that, when the number of *n*_2_ units is fixed, BER generally drops with the increase in *n*_1_, although there are slight fluctuations. Likewise, BER curved surfaces show the same trend regarding the increase in *n*_2_, vertically viewed on the axis of coordinates. The results imply that the BER performance can be improved by increasing *n*_1_ or *n*_2_. 

Furthermore, we compared the complexity of different TLD schemes in Table 1, where the *n*_1_ and *n*_2_ data pair mark BER reaching the HD-FEC threshold with these hidden layer parameters. In addition, we also listed the complexity of the optimal 2nd Volterra equalizer in the same table, despite its BER not reaching the HD-FEC threshold in the same conditions. The complexity of a typical *M*-layer NN is described as $C_{MLP}=n_{0}n_{1}+n_{1}n_{2}+n_{2}n_{M}$ [44], where $n_{i}$ denotes the number of nodes in the *i-th* layer, with $n_{0}$ and $n_{M\;}$ defining the numbers of nodes in the input layer and output layer, respectively. Moreover, for a traditional Volterra equalizer, the total number of multiplications offers the following complexity measure: $C\left(N,L\right)=\sum_{n=1}^{N}\frac{\left(L-1+n\right)!}{\left(n-1\right)!\left(L-1\right)!}$ [45], with *N* representing the Volterra order and *L* denoting the memory length. According to Table 1, the complexity of the optimal 2nd Volterra with 321 taps is close to that of TLD-ROS equalizers, although its effect on nonlinear compensation is far less than TLD-ROS equalizers. Therefore, it can further prove the superiority of a TLD-ROS_QAM_ equalizer, because it reaches the HD-FEC threshold with the simplest structure. 

According to our test, the Adam optimizer was used, the batch size was 128, and the network required 30 epochs to converge. In our proposed TLD-ROS_QAM_ equalization scheme, the trained neural network (NN) received plenty of selective oversampled data on which to train its few classification models. This implies that these samples in the minority class receive more training epochs than those in the majority class. Instead, the computation burden does not increase in our well-trained neural network. Furthermore, the complexity can be decreased by decreasing the amount of neural cells, and we see a complexity reduction of 45.6% for TLD-ROS_QAM_ compared with a typical TLD equalizer without balancing ROS.

## 5. Conclusions

In this paper, we proposed a novel equalization scheme, applying two-lane DNN combined with balancing ROS to improve the overall W-band long-range wireless transmission performance. Moreover, its effectiveness was verified by PS-16QAM 4.6 km ROF delivery experiment results. In our proposed nonlinear equalization scheme, we employed ROS before TLD classification to avoid the class imbalance problem introduced by the PS technique. The experiment demonstrates that the ROS algorithm could enhance system performance without increasing the complexity of the prediction phase. In terms of the receiver sensitivity, the PS-16QAM signal using TLD-ROS_QAM_ equalizers showed a 1dB gain over typical NN equalizers without balancing ROS when the BER reached the HD-FEC threshold of 3.8 × 10^−3^. We compared the performance regarding training accuracy, sample size requirement, and network structure complexity between the TLD, TLD-ROS_I/Q_, and TLD-ROS_QAM_ schemes. We found that there was less demand for sample capacity and network structure depth by using TLD-ROS_QAM_, which enables leaning imbalance mitigation. Moreover, we observed a complexity reduction of 45.6% for TLD-ROS compared with a typical TLD equalizer without balancing ROS. Considering the actual wireless physical layer and its requirements, there is much to be gained from the joint use of deep learning and the pre-processing technique of balancing data.

## Figures and Tables

**Figure 1 sensors-23-04618-f001:**
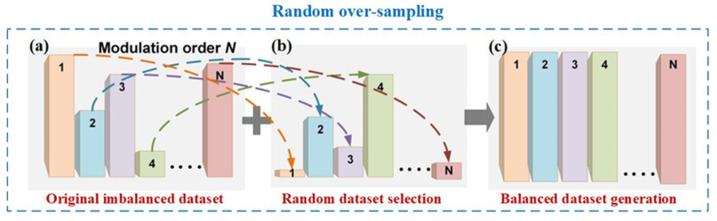
The principle of random oversampling and the corresponding target output of (**a**) the original imbalanced dataset, (**b**) the random oversampled dataset, and (**c**) the balanced dataset.

**Figure 2 sensors-23-04618-f002:**
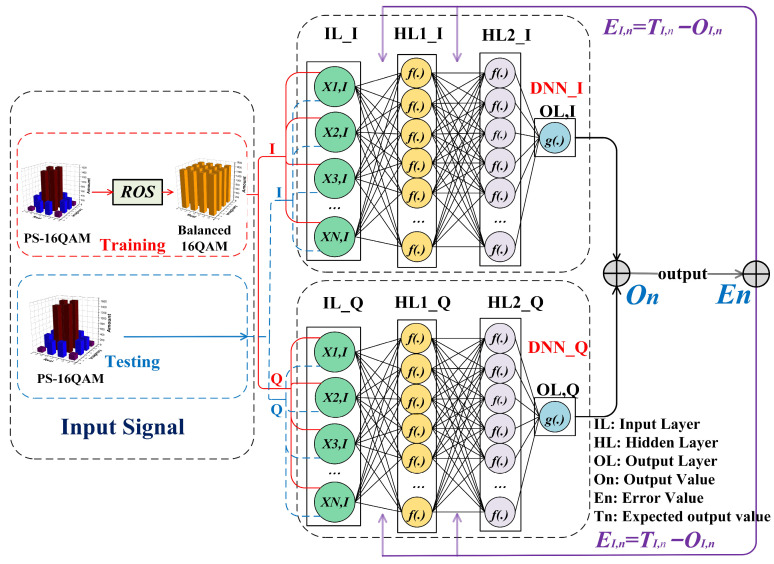
The schematic diagram of a two-lane DNN (TLD) equalizer combined with ROS. IL: input layer. HL: hidden layer. OL: output layer. *O_n_*: output value. *E_n_*: error value. *T_n_*: expected output value.

**Figure 3 sensors-23-04618-f003:**
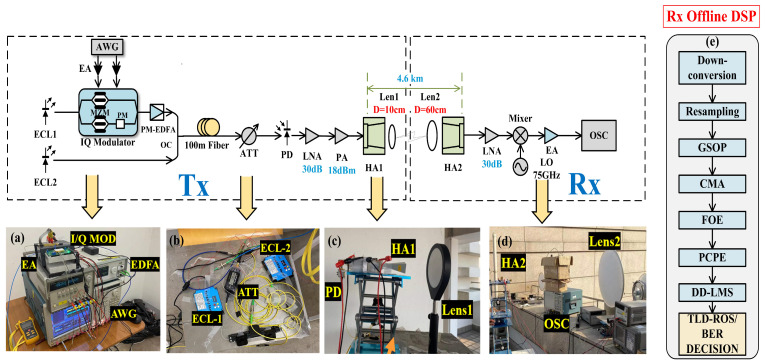
The ROF experimental setup for W-band PS-16QAM RoF delivery over 4.6 km and photos of (**a**) the I/Q modulator, EA, PM-EDFA, and AWG; (**b**) ECLs and the ATT; (**c**) the PD, HA, and lens at the Tx-side; (**d**) the PD, HA, and lens at the Rx-side; (**e**) the block diagram of Rx DSP.

**Figure 4 sensors-23-04618-f004:**
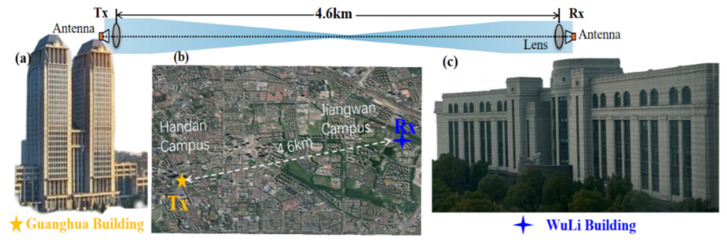
(**a**) Experimental photo of the transmitter; (**b**) satellite map of W-band PS-16QAM mm-wave signal long distance radio-over-fiber delivery on Fudan campus; (**c**) experimental photo of the receiver.

**Figure 5 sensors-23-04618-f005:**
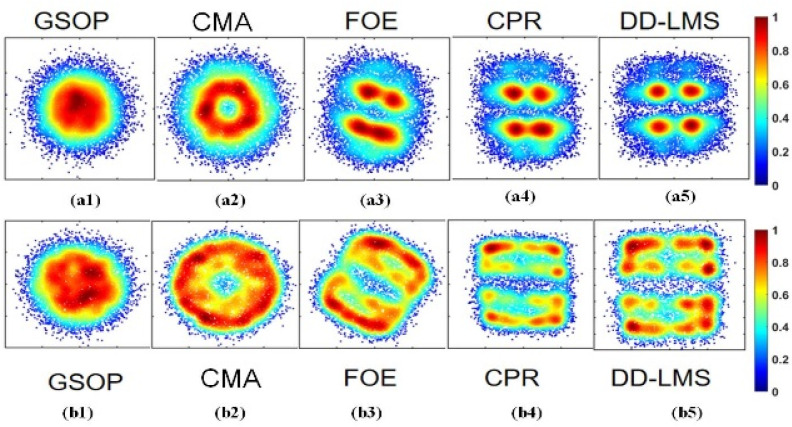
The density distribution of PS-16QAM constellation diagrams after (**a1**) GSOP; (**a2**) CMA; (**a3**) FOE; (**a4**) CPR; and (**a5**) DD-LMS, and the density distribution of 16QAM constellation diagrams after (**b1**) GSOP; (**b2**) CMA; (**b3**) FOE; (**b4**) CPR; and (**b5**) DD-LMS.

**Figure 6 sensors-23-04618-f006:**
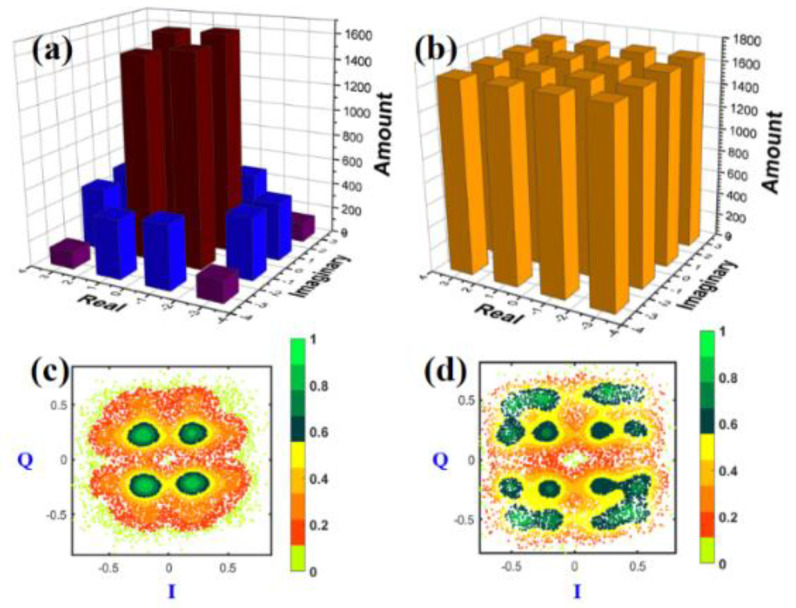
Transmitted PS−16QAM distribution amount (**a**) before ROS and (**b**) after ROS; and the density distribution of received PS−16QAM constellation diagrams (**c**) before ROS, and (**d**) after ROS.

**Figure 7 sensors-23-04618-f007:**
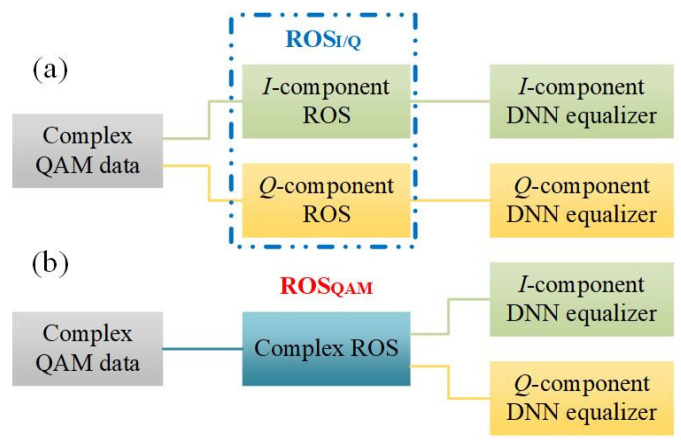
The scheme of TLD equalizers combined with ROS. (**a**) TLD equalizer combined with ROS_I/Q_; (**b**) TLD equalizer combined with ROS_QAM_.

**Figure 8 sensors-23-04618-f008:**
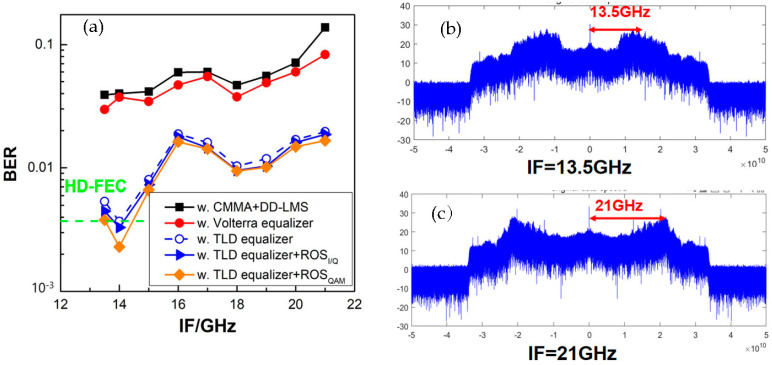
(**a**) The BER performance vs. the intermediate frequency after down−conversion for 10−Gbaud PS−16QAM signals, and the electrical spectrum when the received IF was (**b**) 13.5 GHz and (**c**) 21 GHz.

**Figure 9 sensors-23-04618-f009:**
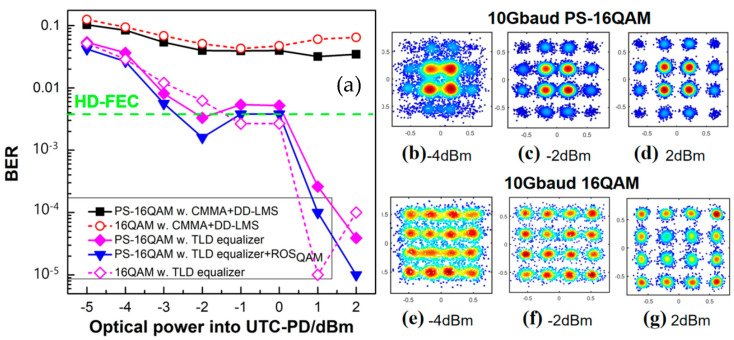
(**a**) BER performance vs. the input optical power of PD for 88.5 GHz PS−16QAM signals and 88.5 GHz 16QAM signals at 10−Gbaud by employing a 45−tap CMMA equalizer, combined with 301−tap DD−LMS, nonlinear TLD equalizers, and TLD equalizers combined with ROS, respectively. The recovered constellation diagrams of the PS−16QAM signal using TLD equalizers when the input optical power of PD was (**b**) −4 dBm; (**c**) −2 dBm; and (**d**) 2 dBm; and that of the 16QAM signal when the input optical power of PD was (**e**) −4 dBm; (**f**) −2 dBm; and (**g**) 2 dBm.

**Figure 10 sensors-23-04618-f010:**
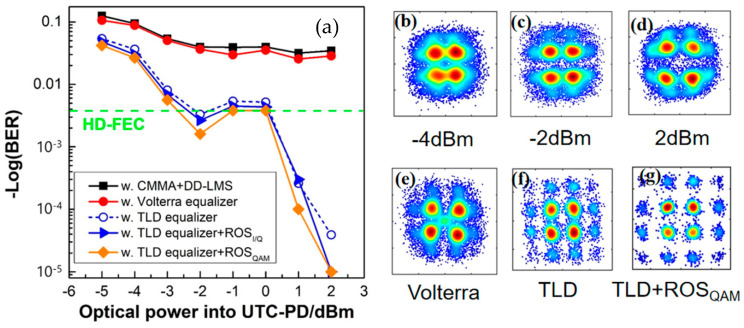
(**a**) The BER performance vs. the input optical power of PD for 88.5 GHz PS−16QAM signals at 10−Gbaud using traditional DSP algorithms, Volterra equalizers, and different TLD−ROS nonlinear equalizer schemes with MSE loss functions. The recovered constellation diagrams of the PS−16QAM signal after DD−LMS when the input optical power of PD was (**b**) −4 dBm; (**c**) −2 dBm; (**d**) 2 dBm. Different constellation performances of the PS-16QAM signal when the input optical power of PD was fixed at 0 dBm, after (**e**) the Volterra equalizer; (**f**) the TLD equalizer; and (**g**) the TLD + ROSQAM scheme.

**Figure 11 sensors-23-04618-f011:**
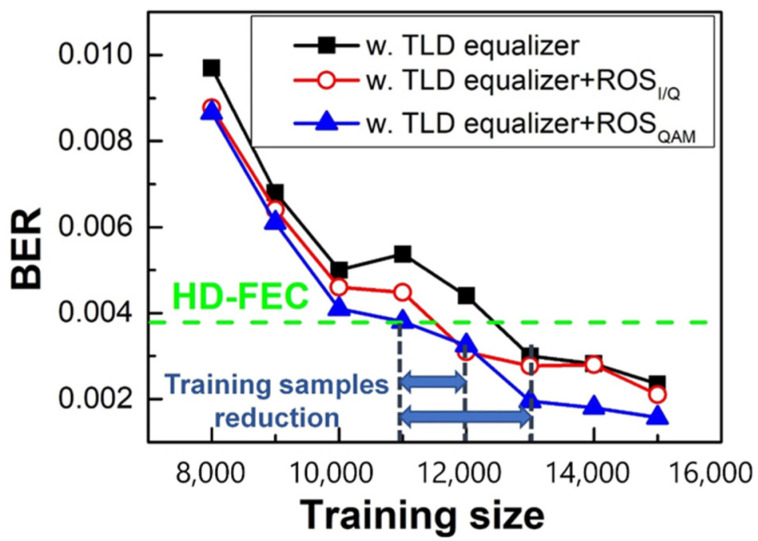
The corresponding BER vs. the training data size for 10Gbaud PS-16QAM signals.

**Figure 12 sensors-23-04618-f012:**
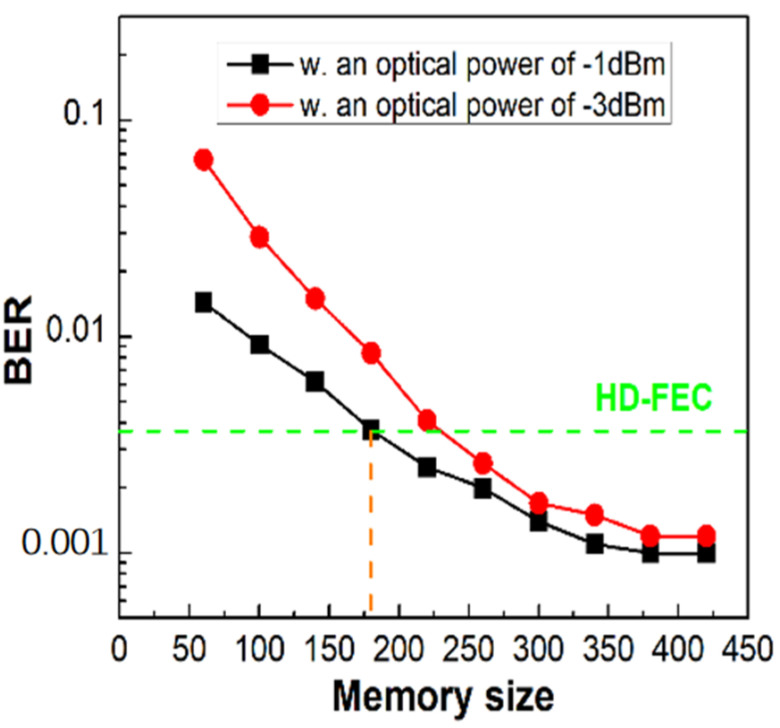
The BER performances of PS-16QAM signals with different levels of optical power vs. the memory size of the NN model.

**Figure 13 sensors-23-04618-f013:**
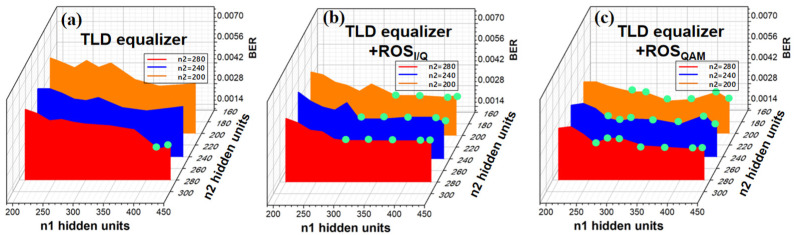
The BER performance for PS-16QAM signals vs. the neural cells *n*_1_ in hidden layer 1 and *n*_2_ in hidden layer 2 when the optical power into PD was −1 dBm, for (**a**) the TLD equalizer, (**b**) the TLD equalizer combined with ROS_I/Q_, and (**c**) the TLD equalizer combined with ROS_QAM_.

**Table 1 sensors-23-04618-t001:** Network parameters of various TLD schemes.

Type	Training Size	*n*_1_/Tap	*n*_2_/Tap	Minimum Multiplication	Complexity Reduction
TLD	13,000	410	280	377,760	0%
TLD-ROS_I/Q_	12,000	320	200	243,600	35.5%
TLD-ROS_QAM_	11,000	270	200	205,600	45.6%
2nd Volterra	13,000	321	321	207,366	45.1%

## Data Availability

The raw/processed data required to reproduce these findings cannot be shared at this time as the data also form part of an ongoing study.

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
