# Peer review of "Two-Lane DNN Equalizer Using Balanced Random-Oversampling for W-Band PS-16QAM RoF Delivery over 4.6 km"

_sensors, 2023, doi:10.3390/s23104618_

Round 1
Reviewer 1 Report
In this research, the authors propose a two-lane DNN (TLD) equalizer utilizing the random over-sampling (ROS) method. They show that W-band wireless transmission system performs better overall when PS at the transmitter and ROS at the receiver are combined. The findings of the authors demonstrate that the 21 TLD-ROS can increase the receiver sensitivity by 1dB in comparison to the conventional TLD without ROS.
The paper is fine except for a few minor comments:
1) The legend for Fig. 13 is missing.
2) What is the reason for the conclusion given in Line 409?
3) In Lines 425-426, the authors mention Adam Optimizer. Did the authors try other optimizers? What was the performance with other optimizers and why Adam Optimizer was used?
Reviewer 2 Report
The paper by L. Zeng et al is devoted to carrying out a novel equalization scheme with applying two-lane Deep Neural Network (DNN) combined with balancing the random over-sampling (ROS) technique to improve the overall W-band long-range wireless transmission performance. As usually, the probabilistic shaping (PS) technique is introduced in order to provide the further enlarge the capacity of channel. The authors have experimentally realized the single-channel 10-Gbaud W-band PS-16QAM transmission over a 100 m optical link and 4.6-km wireless link. The corresponding effectiveness has been proved. The authors have shown that in terms of the receiver sensitivity the PS-16QAM signal using the author's equalizers type shows a 1dB gain over typical neural networks equalizers without balancing ROS when the BER reaches the HD-FEC threshold of 3.8×10-3 . Therefore, the authors have shown that their scheme has a better performance ( in good training accuracy, less training size demand, and computation complexity), compared with the traditional neural networks equalization ones.
The paper by L. Zeng et al has the scientific merits, is of a great theoretical and practical importance, and is definitely recommended for publication in the journal "Sensors" (MDPI).
The only minor points are as follows:
i) In section 2, the authors describe the A two-lane DNN (TLD) equalizer used. A detailed description of the neural networks and training scheme is also presented.or example, the authors use a fairly simple, low-layer neural network. As nonlinear active function between hidden layer the function (5) is used. The readers could have a question regarding the motivation for choosing network parameters. It would be desirable to explain the influence of the choice of network parameters on the final results (at least briefly within a few lines).
ii). The authors use a lot of abbreviations and do not decipher their meaning everywhere, which will cause certain difficulties for readers in perceiving the results of the work. It would be highly desirable to check that all quantities in formulas, as well as all abbreviations, are defined and described.
iii) In order to take the possible questions of the readers into account, it makes sense to expand the list of references and add a few references on the known monographies (textbooks) on neural networks and deep learning, for example, as follows:
1., A. V. Glushkov, A.A. Svinarenko, A. V. Loboda, Theory of Neural Networks on Basis of Photon Echo and Its Program Realization. TEC, Odessa, 2004.
2.,Optical telecommunication systems. Ed. V.N. Gordienko. Hotline-Telecom, Moscow,2011.
3.,D Grauoe, Deep Learning Neural Networks Design and Case Studies. World Scientific, 2016.
4. ,I. Goodfellow, Y. Bengio, and A. Courville, Deep Learning, the MIT Press, 2016.
5., C.C.Aggarwal, Neural Networks and Deep Learning: A Textbook. Springer Cham, 2018 ( https://doi.org/10.1007/978-3-319-94463-0)
and others at the discretion of the authors.
Thanks for the revisions.
